# COVID-19 Pandemic Impact on the Maternal Mortality in Kazakhstan and Comparison with the Countries in Central Asia

**DOI:** 10.3390/ijerph20032184

**Published:** 2023-01-25

**Authors:** Olzhas Zhamantayev, Gaukhar Kayupova, Karina Nukeshtayeva, Nurbek Yerdessov, Zhanerke Bolatova, Anar Turmukhambetova

**Affiliations:** School of Public Health, Karaganda Medical University, Gogol Street 40, Karaganda 100008, Kazakhstan

**Keywords:** maternal mortality, COVID-19, Central Asian countries, Kazakhstan

## Abstract

Maternal mortality ratio is one of the sensitive indicators that can characterize the performance of healthcare systems. In our study we aimed to compare the maternal mortality ratio in the Central Asia region from 2000 to 2020, determine its trends and evaluate the association between the maternal mortality ratio and Central Asia countries’ total health expenditures. We also compared the maternal mortality causes before and during the pandemic in Kazakhstan. The data were derived from the public statistical collections of each Central Asian country. During the pre-pandemic period, Central Asian nations had a downward trend of maternal mortality. Maternal mortality ratio in Central Asian countries decreased by 38% from 47.3 per 100,000 live births in 2000 to 29.5 per 100,000 live births in 2020. Except for Uzbekistan, where this indicator decreased, all Central Asian countries experienced a sharp increase in maternal mortality ratio in 2020. The proportion of indirect causes of maternal deaths in Kazakhstan reached 76.3% in 2020. There is an association between the maternal mortality ratio in Central Asian countries and their total health expenditures expressed in national currency units (*r* max = −0.89 and min = −0.66, *p* < 0.01). The study revealed an issue in the health data availability and accessibility for research in the region. The findings suggest that there must be additional efforts from the local authorities to enhance the preparedness of Central Asian healthcare systems for the new public health challenges and to improve health data accessibility.

## 1. Introduction

The COVID-19 pandemic has had a significant impact on healthcare systems around the world. Various findings highlight that restrictions, quarantines, and overwhelming pressure on healthcare systems have caused some dramatic changes in different health performance indicators, such as healthcare utilization and, excess mortality, etc. The pandemic has disrupted essential healthcare services, including maternal and newborn care, leading to an increase in maternal mortality (MM) [1,2]. Maternal mortality ratio (MMR) is a key indicator that can characterize the overall state of healthcare systems and it can be used to assess and compare countries’ socioeconomic performance and healthcare effectiveness [3,4].

There are several middle-income countries such as Kazakhstan, Uzbekistan, Kyrgyzstan, Tajikistan, and Turkmenistan in the Central Asia (CA) region that share common issues and trends in economy and politics [5,6]. Some of them have experienced similar obstacles in healthcare during the COVID-19 pandemic, such as a lack of affordable PCR testing or management of chronic diseases at the beginning of the pandemic [7,8]. In addition, Kazakhstan, Uzbekistan, and Kyrgyzstan have a relatively low share (2.5–5%) of GDP spent on healthcare, whereas many high-income countries have much more significant amounts (more than 7%) of GDP on healthcare [9,10].

WHO reports that 94% of all maternal deaths occur in low and lower-middle-income countries, and MMR dropped by about 38% worldwide between 2000 and 2017 [11,12]. Moreover, several years before the pandemic about 86% of all maternal mortality (MM) cases were associated with direct obstetric causes (maternal hemorrhage, maternal hypertensive disorders, and other maternal disorders) [13,14]. Conversely, three-fifths of maternal deaths in Georgia from 2014 to 2017 were related to indirect causes, with infectious diseases being the most frequent one and almost half of the maternal deaths were linked to pre-existing medical conditions [15]. The UN’s Sustainable Development Goal 3.1 aims to reduce the global MMR to less than 70 per 100,000 live births. Meanwhile, this promising goal can be compromised by the fact that only 84% of births in the world were assisted by skilled health professionals (doctors, nurses, and midwives) during 2015–2021. This indicator has reached 95% in Tajikistan, 99.8% in Kyrgyzstan, 99.9% in Kazakhstan, and 100% in Uzbekistan [16,17]. In comparison to these countries, the level of births attended by skilled health personnel was 96.2% in Germany, 98% in Canada, and some middle and high-income African countries had only 90% in 2018 [17,18]. There are studies that found an association between the MMR and government health expenditures or funds allocation on providing sufficient health coverage. The increased health expenditure potentially reduces maternal and infant mortality across lower- and middle-income countries. Lower-middle-income countries reported significantly higher rates of maternal mortality than high-income countries [19,20,21,22]

Scientific evidence on the influence of the COVID-19 pandemic on maternal mortality and pregnancy course varies among the different studies. Chmielewska et al. identified significant increases in stillbirth and maternal death during pandemic versus before the pandemic [2]. Some researchers concluded that COVID-19 infection did not significantly affect the course of pregnancy or pregnancy outcomes [23,24,25,26], and unusual frequency of pregnancy-related complications due to SARS-CoV-2 infection was also not revealed [27]. However, pregnant women with COVID-19 may have more adverse health outcomes and experience socio-economic issues [28,29]. SARS-CoV-2 infection among pregnant and postpartum women was associated with an increased risk for a serious morbidity related to hypertensive disorders of pregnancy, postpartum hemorrhage, or death [30,31]. There are findings showing that maternal deaths have increased approximately five times among hospitalized patients during the COVID-19 Delta wave and SARS-CoV-2 infection in pregnancy is associated with reduced risk of complications during the omicron-dominant period compared with the delta-dominant period [32,33]. Several studies identified an association of COVID-19 infection with higher rates of cesarean section in pregnant women and their mortality [34,35,36]. Pre-existing comorbidities such as obesity, diabetes, asthma, and advanced maternal age played a role in the increased complications risk of COVID-19 and maternal mortality [37,38,39].

In 2021, Brazil reported 223% more maternal deaths caused by COVID-19 than in 2020. Authors identified delays in diagnosis, hospitalization, and providing intensive care as three main barriers that impede effective healthcare responses to pregnant women with COVID-19 [40]. The projected MMR in Mexico for 2020 (before the pandemic) was 29.5 per 100,000 live births, yet in fact it was 42.4 that year [41].

Our study aims to compare the MMR situation in CA countries, assess its trends from 2000 to 2020, analyze the association between MMR and total health expenditures (THE) expressed in national currency units in CA countries, and compare the causes of MM before and during the COVID-19 pandemic in Kazakhstan.

## 2. Materials and Methods

### 2.1. Data Sources

MMR is estimated as the number of maternal deaths per 100,000 live births. It is defined as the death of women who are pregnant or who are within 42 days of the termination of pregnancy, regardless of the length and location of the pregnancy, from any cause related to or aggravated by the pregnancy or its management, but not from an accident or accidental causes [42].

We conducted a retrospective study of the data set of four CA countries, including Uzbekistan, Kyrgyz Republic, Tajikistan, and Kazakhstan, from the official state statistical profiles and reports of each country [43,44,45,46]. We excluded Turkmenistan data from our analysis due to data unavailability.

The annual statistical report *“Population health and healthcare organizations’ performance in the Republic of Kazakhstan”* was used to derive data about the MMR, pregnancy complications, causes of maternal deaths, obstetrician-gynecologists’ density, and maternal beds per 1000 born in Kazakhstan from 2000 to 2020. The health data collection policy in Kazakhstan complies with international recommendations to ensure the comparability of statistical data (measurement, recording, classification, sources linking and reporting) [47].

Maternal mortality statistics in Kyrgyzstan are maintained by the Department of Social and Environmental Statistics of the National Statistical Committee, and data are collected and processed in cooperation with the Ministry of Health and the Ministry of Social Care. The competent authorities carry out control of the received data, prepare and verify summary information. Maternal mortality statistics are published in the annual reports of the National Statistical Committee.

The collection of data on maternal mortality in Uzbekistan and Tajikistan is carried out by the Confidential Study Report on Maternal Deaths by United Nations Population Fund (UNFPA) in Uzbekistan and the Statistics Agency of Tajikistan. The review of compiled health statistics, including maternal mortality, is carried out in accordance with international recommendations by the competent authorities and published in annual health statistical reports [47].

The Global Health Expenditures Database was used to derive data about total health expenditures (THE) of CA countries in national currency units (NCU) [48].

### 2.2. Maternal Mortality Registration Process in CA Countries

According to the Order of the Minister of Health of the Republic of Kazakhstan, the heads of local health facilities, from the moment of MM case registration, are required to notify the authorized regional body within two hours by telephone. Then, within 24 h, they report it to the Ministry of Health and its territorial divisions with the available results (medical records, clinical, laboratory, and instrumental examination). The “protocol of post-mortem examination” must be submitted to the territorial divisions within seven working days from the date of MM registration. Control over the registration and accounting of MM is carried out by the regional head of the health department. The Republican E-Health Center provides monthly summary information on MM cases by the eighth day of each month following the reporting month, indicating the final diagnosis and the ICD-10 code (rules for the provision of information [49]).

Pathological anatomical autopsies are performed in all cases of MM by physicians specializing in “pathological anatomy (adult, pediatric) with pathohistological examination of the sectional material, at the pathology bureaus, centralized pathology departments within 24 h after the death”. Upon completion of the entire pathologic examination, all maternal deaths are subject to a clinicopathologic anatomic review. The pathological and anatomical diagnosis shall be made in accordance with the ICD. The Department of the Ministry of Health of Kazakhstan ensures the timely monitoring of MM cases along the route of pregnant women/parturient women/puerperal/women after the termination of pregnancy if there are legal grounds in accordance with the legislation of the Republic of Kazakhstan. In addition, the department provides information on the results of inspections to the authorized body for taking urgent measures to reduce maternal mortality [50].

The Ministry of Health of Kyrgyzstan collects up-to-date information on women who died from complications of pregnancy, childbirth, and the postpartum period on a monthly basis and submits it to the national statistical committee, where the fact of registration of these cases in the registry offices is being clarified. “The signal certificate of death of a pregnant woman/a woman in labor/a puerperal woman” is filled in in all healthcare organizations (regardless of whether this organization provides obstetric services or not). It is filled in by entering data in the appropriate paragraphs or rounding off the corresponding codes. In the case of death of a woman in a hospital, it is indicated during what time from the moment of admission to the hospital the death occurred. It reflects information about the medical staff who took delivery. Other important conditions that contribute to death, but are not associated with the disease or pathological condition that led to it, are also coded. The certificate is transferred within 3 days to the regional medical information center, where data will be entered into the MM register. A head of a healthcare organization is responsible for the timeliness of the submitted data, the quality of filling in the certificates, and the completeness and timely entry of information into the database [51].

Information on the process of registering a case of maternal death in Tajikistan and Uzbekistan is not presented in this study due to its unavailability.

### 2.3. Data Analysis

We used linear regression analysis to identify statistically significant trends of MMR in CA countries from 2000 to 2020 in this study. Pearson correlation analysis was used to assess the associations between the observed national MMRs and the CA countries’ total health expenditures in NCU. R-studio soft version 1.2.5033, Posit, PBC, Vienna, Austria was used for the statistical analysis in this study, and a two-sided *p*-value of <0.01 was considered significant. We used the following code in R:

Correlation <- cor(data$x, data$y, method = ‘pearson’)

Checking the results: >correlation

## 3. Results

### 3.1. MM Situation in CA Countries and Its Trends from 2000 to 2020

In general, Figure 1 shows a general downward trend in all four countries, but fluctuation year-on-year from 2000 to 2020. The highest level of MMR in the CA region was 47.3 per 100,000 live births (95% CI: 29.0, 65.5) in 2000, and the lowest level of MMR was 20.4 per 100,000 live births (95% CI: 12.5, 28.4) in 2019 (Appendix A, Table A1). Overall, MMR in CA countries decreased by 38% from 47.3 (95% CI: 29.01, 65.5) per 100,000 live births in 2000 to 29.5 (95% CI: 15.8, 43.1) per 100,000 live births in 2020. In 2020, the MMR in CA countries increased by 44% in comparison to 2019. It is worth noting that throughout this period, Kyrgyzstan had the highest MMR among the CA countries. In addition, the fluctuations of MMR have taken place in the previous periods of time. There were increases between 2001–2002, 2004–2005, 2008–2009 and 2013–2014 which were comparable in scale to the increase between 2019 and–2020.

Overall, there is a 20-year decreasing trend of MMR in Kyrgyzstan. Linear regression analysis results of a 20-year period of MMR decline in Kyrgyzstan showed a coefficient of −1.31 (95% CI: −1.91, −0.70, *p*-value < 0.001) and an intercept of 2671.28 (*p* < 0.001). The trend of the yearly MMR in Kyrgyzstan was −1.31, which indicates an average annual decline of 1.31 maternal deaths per 100,000 live births per year. The peak of MMR was in 2009, reaching 63.5 per 100,000 live births. From 2011, this indicator tended to decline year after year from 54.8 (2011) to 24.8 (2019). However, during the pandemic, the MMR there sharply increased up to 36.1 per 100,000 live births, increasing by 45.5% in 2020 in comparison to previous year.

During the last two decades, progress was made in managing maternal deaths in Tajikistan. Similar to Kyrgyzstan, Tajikistan has a downward trend year after year, from 49.6 per 100,000 live births in 2000 to 23.6 in 2019 with some fluctuations in 2005–2006 and 2007–2008. The indicator decreased by 47.5% from 2000 to 2019. During the pandemic, the MMR slightly increased by 13.1% in 2020, reaching 26.7 per 100,000 live births. Linear regression analysis results of the 20-year period of the MMR in Tajikistan showed a coefficient of −1.15 (95% CI: −1.44, −0.88, *p*-value < 0.001) and an intercept of 2360.37 (*p* < 0.001). The trend of the yearly MMR in Tajikistan was −1.15, which indicates an average annual decline of 1.20 maternal deaths per 100,000 live births.

According to the national statistics of Uzbekistan, COVID-19 did not affect the MMR. It is one of the few countries where MMR did not increase recently. As COVID-19 hit the world, this indicator in Uzbekistan was lower in 2020 (18.5 per 100,000 live births) than a year before (19.6 per 100,000 live births), decreasing by 6%. The MMR decreased from 33.1 (2000) to 22.4 (2008) and peaked reaching 30.4 in 2009. Since that period, it has been steadily declining, but compared to the developed countries of the world, it is still relatively high. Linear regression analysis results of the 20-year period of the MMR in Uzbekistan showed a coefficient of −0.79 (95% CI: −0.99, −0.61, *p*-value < 0.001) and an intercept of 1628.38 (*p* < 0.001). The trend of the yearly MMR in Uzbekistan was −0.79, which indicates an average annual decline of 0.79 maternal deaths per 100,000 live births.

Kazakhstan has the lowest MMR among CA countries since 2000. Like the countries mentioned above, it also has had a downward trend during the last decade. However, this indicator increased sharply by 2.6 times in 2020 (36.5 per 100,000 live births) compared to 2019, and it showed the highest hand among CA countries that year. It was a record peak in the history of Kazakhstan over the past ten years (Figure 2). Moreover, a sharp increase in maternal deaths in Kazakhstan has leveled with the MMR in Kyrgyzstan.

According to Table 1, the absolute number of THE increased in all CA countries from 2000 to 2019. These values have been increasing in every country since 2015, except for Kyrgyzstan. Pearson correlation analysis showed that the MMR of each CA country was negatively correlated with the absolute number of THE in NCU (*r* = −0.86 for Kazakhstan, *r* = −0.70 for Kyrgyzstan, *r* = −0.66 for Uzbekistan, and *r* = −0.91 for Tajikistan, *p* < 0.01).

### 3.2. The Comparison of the MM Causes before and during the Pandemic in Kazakhstan

The MMR in Kazakhstan decreased by 4.4 times between 2000 (60.9 per 100,000 live births) and 2019 (13.7 per 100,000 live births), followed by a sharp increase of 2.7 times in 2020 (36.5 per 100,000 live births, Figure 1) during the COVID-19 outbreak. Linear regression analysis results of the MMR in Kazakhstan showed a coefficient of −2.13 (95% CI: −2.8, −1.41, *p*-value < 0.001) and an intercept of 4300.89 (*p* < 0.001). The trend of the yearly MMR in Kazakhstan was −2.13, which indicates an average annual decline of 2.13 maternal deaths per 100,000 live births.

In Figure 3, the main causes of maternal mortality are provided. In Kazakhstan, during 2000–2020, the main reasons for maternal mortality were broken down into the following categories according to ICD: indirect causes (O10, O24, O98, O99), hemorrhage (O67, O46, O44, O45, O72), abortion (O03–O07), hypertensive disorders (O10–O16), sepsis (O85), uterine rupture (O71.1), ectopic pregnancy (O00), and other (O21.1, O22, O71, O73, O87, O88, O90).

Abortion and hemorrhage were the leading causes of maternal deaths in 2000. (28% and 21%, respectively). Afterwards, the MM structure changed, with indirect causes becoming the leading cause of death, reaching a peak in 2020 (76% among all causes). There has also been a remarkable decrease in mortality from hemorrhage, abortion, pregnancy hypertensive disorders, and sepsis since 2000 until 2020 (in total there are 18 MM cases due to these causes in 2020 (Figure 3).

The obstetrician-gynecologists’ density per 1000 births (including stillbirths) decreased from 13.9 in 2000 to 7.3 in 2020. The situation is similar with the maternal beds density, there is a significant decrease from 39.5 in 2000, to 15.5 per 1000 born in 2020 (Appendix A, Table A1). It is worth noting that the indicators mentioned above did not show a significant decrease during the coronavirus pandemic, as a decrease was observed throughout the study period.

Another observed trend is a year-by-year steady increase in the number of women registered for prenatal care from 223,515 in 2000 to 435,358 in 2020. Prenatal care coverage, including early registration for up to 12 weeks of pregnancy, was at the critically low level of 66.5% twenty years ago and rose to 83.9% in 2020. Pregnancy complications such as hypertensive disorders and anemia are still the most common in Kazakhstan. The number of pregnancy complications has been declining for 22 years, including the pandemic period (Appendix A, Table A3).

## 4. Discussion

According to the results of our study, the MMR in Kazakhstan and other CA countries has been decreasing annually on average for the last twenty years. In the year of the pandemic there was an increase of MMR in all CA countries, except for Uzbekistan. In Kazakhstan the MMR decreased by 50% over 23 years with indirect causes, pregnancy hypertensive disorders being the leading causes of MM. Overall, the positive trends were a result of an effective policy on maternal care in the region: all countries have embarked on primary health care reforms, nursing training upgrade, and involvement of local communities and NGOs in the development of quality improvement programs in the national health plans. There was also an emphasis on the introduction of the evidence-based medicine and multisectoral approach in all health sectors, including maternal care in the region [52,53].

In the past 10 years, indirect causes have become one of the leading causes of maternal deaths in different parts of the world [14,15]. Moreover, in a pandemic year, every three out of four deaths were due to this cause. Thoma et al. found that MM increased by 33.3% in the United States during the pandemic. Viral, respiratory, and circulatory system diseases were the major causes of MM there [54]. Indirect causes have become the main cause of death among pregnant women and mothers in Kazakhstan since 2012.

The CA countries have one of the highest percentages of births attended by skilled health professionals worldwide, almost reaching 100% [17,18]. According to Amorim et al., in developing countries, as opposed to developed countries, high fertility and limited resources for health services will identify an increased risk of MM due to COVID-19 and they highlight the need for appropriate measures for adequate prenatal and postnatal care [55]. The average total fertility rate (TFR) in Kazakhstan, Kyrgyzstan, Uzbekistan and Tajikistan (3.0) in 2020 was higher than in the world (2.3), and higher than in EU (1.5). There was a steady growth of the fertility rate in three countries (Kazakhstan, Kyrgyzstan, and Uzbekistan) over a twenty-year period 2000–2020 (Appendix A, Table A2) [56].

There are few publications which attempt to assess the quality and completeness of the data sources in the CA region. All countries have eHealth centers that collect and publish annual health statistics information. The legibility of data is considered good in Kazakhstan and Kyrgyzstan [57,58].

There are quality councils in Kyrgyzstan which have little support in terms of tools, statistical benchmarks or local patient-based data for systematic peer review and improvement. United Nations experts identified the need for capacity development support in implementing measures to manage quality, and to raise awareness of the importance of quality management throughout the national statistical system [59,60]. The efficiency of data harvesting and processing could be improved by stimulating electronic data collection in all CA countries [61].

The THE varies across the CA countries. There are increased investments in this field but still the level is incomparable to middle-income countries from other parts of the world. Currently, only Tajikistan has achieved the level recommended by WHO (7% and more of GDP spent on healthcare) [48]. According to studies most of the countries with high excess mortality rates due to COVID-19 have insufficient investments in healthcare systems [22,39,41,62]. We believe that the limited financing of maternal care could have had an influence on the increased MMR in some CA countries, especially during the COVID-19 pandemic.

### Limitations

Moreover, some statistical indicators regarding MM, such as causes of MM, obstetricians’, and maternal beds’ density per 1000 born, and prenatal care coverage were not available in CA countries’ state statistical sources. The quality of surveillance systems and data quality may differ across the selected countries.

Furthermore, due to limited data on the monetary resources expressed in NCUs it is not clear how much is exactly allocated for maternal care.

Nowadays limited investments in data systems of some CA countries compromise the evaluations of maternal mortality there.

## 5. Conclusions

Preventing MM and morbidity has become a global goal. The MMR is the indicator, which sums up the efforts on managing the medical, social, economic, environmental, cultural and organizational factors. In general, the MMR in CA countries has been declining on average for the last decades. The fluctuations of MMR have taken place in the previous periods of time. In Kazakhstan, the rate increased by two and a half times in the year of the pandemic compared to the previous year, although there had been a downward trend since 2000. Indirect causes have become the main cause of maternal deaths since 2012 in Kazakhstan, and in 2020, their share grew by 19%. The MMR of every CA country taken for analysis was negatively correlated with the absolute number of THE expressed in NCU. The COVID-19 pandemic can be the reason for the recent sharp increase in MM; however, more time and data are needed to draw any specific conclusions about its impact on maternal mortality. Meanwhile, it exposed problems in CA countries’ healthcare systems such as lack of data availability and low investments in the healthcare sector.

## Figures and Tables

**Figure 1 ijerph-20-02184-f001:**
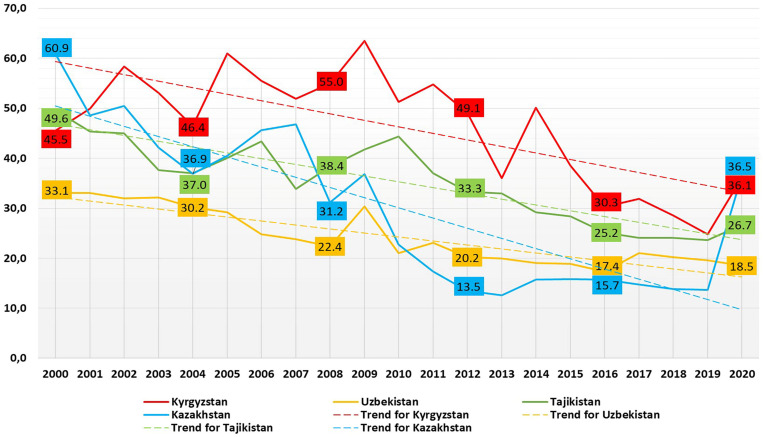
Maternal mortality ratio in CA countries, per 100,000 live births, 2000–2020.

**Figure 2 ijerph-20-02184-f002:**
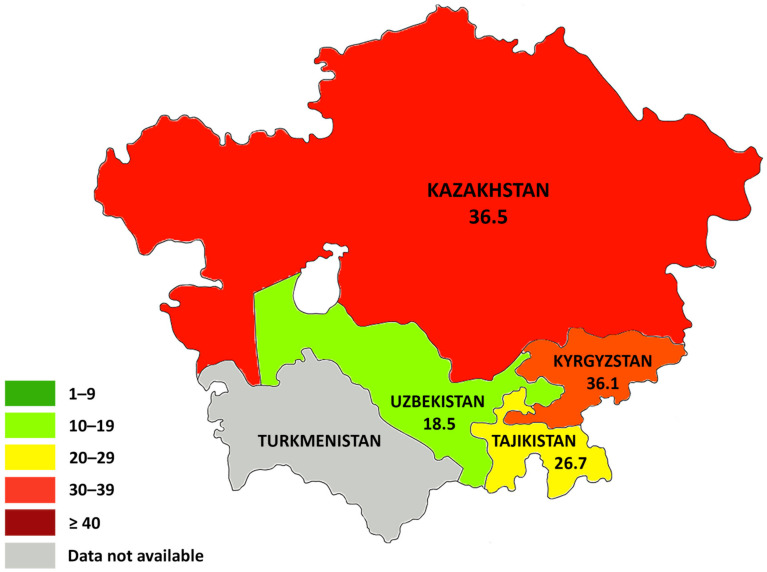
Maternal mortality in CA countries per 100,000 live births in 2020.

**Figure 3 ijerph-20-02184-f003:**
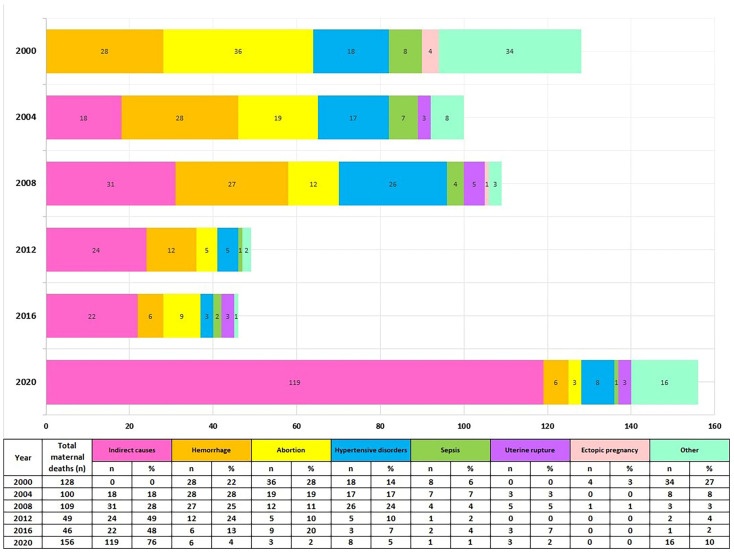
MMR structure by the cause of death in Kazakhstan from 2000 to 2020.

**Table 1 ijerph-20-02184-t001:** Total health expenditures of Central Asia countries in national currency units from 2000 to 2019 in millions.

Year	Kazakhstan (Tenge)	Kyrgyzstan (Som)	Uzbekistan (Sum)	Tajikistan (Somoni)
2000	108.164	2.885	209.474	77
2001	112.827	3.160	323.361	115
2002	136.505	3.461	480.187	148
2003	171.720	4.885	639.818	224
2004	233.812	5.852	744.062	296
2005	296.164	7.517	954.605	375
2006	347.088	9.459	1236.557	467
2007	347.349	9.837	1605.195	708
2008	489.538	12.267	2226.225	1041
2009	595.121	13.882	2949.005	1208
2010	596.963	15.316	3866.230	1417
2011	734.988	20.326	4986.864	1762
2012	942.012	26.420	6561.890	2163
2013	958.606	29.090	8183.944	2647
2014	1180.231	29.482	8278.274	3040
2015	1243.087	30.777	10,483.103	3343
2016	1607.520	30.545	12,043.255	3815
2017	1659.885	32.815	15,359.188	4416
2018	1741.988	28.528	21,563.590	4984
2019	1938.192	27.826	28,753.029	5496
*r*	−0.86	−0.70	−0.66	−0.89
*p*	<0.001	<0.001	<0.01	<0.001

## Data Availability

Not applicable.

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
