# Peer review of "COVID-19 Pandemic Impact on the Maternal Mortality in Kazakhstan and Comparison with the Countries in Central Asia"

_ijerph, 2023, doi:10.3390/ijerph20032184_

Round 1
Reviewer 1 Report (Previous Reviewer 3)
I think this article is now publishable. I am ok with the changes.
Author Response
The authors thank the reviewers for the time and efforts spent to improve the manuscript. All the comments are really appreciated. All suggestions were very insightful and inspirational.
We hope that the revised manuscript may be contributing to the journal’s interests and values.

Reviewer 2 Report (New Reviewer)
Many thanks for giving me the opportunity to review this paper. There is relatively little data on maternal mortality published from this region, and this paper fills a needed gap. However, currently the manuscript is missing an explanation of important details that I strongly recommend the authors expand upon before publication. I have divided my comments into major and minor accordingly.
MAJOR COMMENTS
My primary comment is that the authors need to add considerably more information about the source of data used in this paper. Section 2.1 is dedicated to describing the data sources, but the information here is currently too vague/high-level to allow the figures to be adequately interpreted. The authors state several times that the data 'complies with international recommendations' but guidance in this area is not standardised or 'one size fits all'. The citation provided is [43]: however, no country in the world follows every recommendation or best practice provided in this document - and it is therefore unclear to me which recommendations it follows and which is does not. For example, what are the data quality checks in place to ensure completeness and correct classification of material deaths, what data linkage, triangulation or cross-checking takes place, who certifies the cause of death and what training do they have for this? These are the sort of details that need to be added to strengthen this section.
My other major comment relates to providing greater clarity on the classification of the causes of maternal deaths. The process for this would benefit from being described in greater detail (who, when, process etc) as above. In addition, the specifics of the groups used are currently not clear to me as they are similar to but not the same as the ICD. Within the ICD, ectopic pregnancy (O00) falls within Group 1, Pregnancy with abortive outcome; the various codes for uterine rupture fall within Group 3, Obstetric haemorrhage; but in both these cases these groups also appear separately. It's unclear what 'Other' would include - there's an important difference between cause unknown and another cause such as Complications of anaesthesia. I'd suggest adding a few additional sentences to methods explaining this classification, and including a table as an appendix providing a breakdown of how the ICD codes were grouped for each category.
The authors need to justify and explain their analysis choice of simple linear regression to analyse the data - particularly given situation that authors seem to consider 2020 to be different from earlier years.
MINOR COMMENTS
If possible, given the links the authors make to the COVID-19 pandemic, it would be helpful to further sub-divide the column on 'Indirect causes' in Figure 3.
MMR is presented but no absolute numbers and no uncertainty or confidence intervals. Ideally these should be added, eg to Figure 1 and the text. Such details are currently only provided for trends within countries yet the text makes statements such as KAZ having lowest MMR of all the countries, which cannot currently be assessed by the reader.
Changes in MMR may not only be due to changes in risk of death but also due to changes in fertility - I would recommend that this this should be expanded upon in the discussion.
Author Response
Response to Reviewers
The authors thank the reviewers for the time and efforts spent to improve the manuscript. All the comments are really appreciated. We agree in general with all points raised by the reviewers and have revised the manuscript considering the suggestions. All of them were very insightful and inspirational.
We hope that the revised manuscript may be contributing to the journal’s interests and values.
The text below follows all the comments from the reviewers. The reviewer´s text is in black and authors´ replies are in red (same as updates and changes in the revised manuscript).
Response to Reviewer 2:
MAJOR COMMENTS
- My primary comment is that the authors need to add considerably more information about the source of data used in this paper. Section 2.1 is dedicated to describing the data sources, but the information here is currently too vague/high-level to allow the figures to be adequately interpreted. The authors state several times that the data 'complies with international recommendations' but guidance in this area is not standardised or 'one size fits all'. The citation provided is [43]: however, no country in the world follows every recommendation or best practice provided in this document - and it is therefore unclear to me which recommendations it follows and which is does not. For example, what are the data quality checks in place to ensure completeness and correct classification of material deaths, what data linkage, triangulation or cross-checking takes place, who certifies the cause of death and what training do they have for this? These are the sort of details that need to be added to strengthen this section.
Thanks for your valuable comment. We have carried out work on this issue, and please find our additions in the "Methods" section, unfortunately, we still have not found much information from Uzbekistan and Tajikistan.
- My other major comment relates to providing greater clarity on the classification of the causes of maternal deaths. The process for this would benefit from being described in greater detail (who, when, process etc) as above. In addition, the specifics of the groups used are currently not clear to me as they are similar to but not the same as the ICD. Within the ICD, ectopic pregnancy (O00) falls within Group 1, Pregnancy with abortive outcome; the various codes for uterine rupture fall within Group 3, Obstetric haemorrhage; but in both these cases these groups also appear separately. It's unclear what 'Other' would include - there's an important difference between cause unknown and another cause such as Complications of anaesthesia. I'd suggest adding a few additional sentences to methods explaining this classification, and including a table as an appendix providing a breakdown of how the ICD codes were grouped for each category.
Thank you for your important comment. Pathological anatomical autopsies are performed in all cases of maternal mortality by physicians specializing in "pathological anatomy (adult, pediatric) with pathohistological examination of the sectional material, at the PAB (pathology bureaus), CPAD (centralized pathology departments) and PAD (pathology departments) within 24 hours after the death.
Upon completion of the entire pathologic examination, all maternal deaths are subject to Clinicopathologic anatomic review.
The pathological and anatomical diagnosis shall be made in accordance with the ICD [Standard of pathological and anatomical diagnostics in the Republic of Kazakhstan. Order of the Minister of Health of the Republic of Kazakhstan dated December 14, 2020].
- The authors need to justify and explain their analysis choice of simple linear regression to analyse the data - particularly given situation that authors seem to consider 2020 to be different from earlier years.
Thank you for spotting this thing. Here we stand on the point explaining that trend monitoring is important for evaluating the impact of public health interventions. Therefore, it is important to find out whether a trend can take place, and if so, whether it is statistically significant. We performed a trend analysis of maternal mortality using linear regression based on the following publications and guidelines that recommend using linear regression to identify statistically significant trends:
https://www.webofscience.com/wos/woscc/full-record/WOS:000473022700010
https://www.webofscience.com/wos/woscc/full-record/WOS:000684217000008
https://onlinelibrary.wiley.com/doi/epdf/10.1111/1467-9892.00171
MINOR COMMENTS
- If possible, given the links the authors make to the COVID-19 pandemic, it would be helpful to further sub-divide the column on 'Indirect causes' in Figure 3.
Thank you for the comment! It would be really interesting to sub-divide the indirect causes, unfortunately we did not have access to the detailed data on that.
- MMR is presented but no absolute numbers and no uncertainty or confidence intervals. Ideally these should be added, eg to Figure 1 and the text. Such details are currently only provided for trends within countries yet the text makes statements such as KAZ having lowest MMR of all the countries, which cannot currently be assessed by the reader.
We totally agree with you. We added following to the “Result” section.
In general, Figure 1 shows a general downward trend in all four countries, but fluctuation year-on-year from 2000 to 2020. The highest level of MMR in the CA region was 47.3 per 100,000 live births (95%CI:29.0, 65.5) in 2000, and the lowest level of MMR was 20.4 per 100,000 live births (95%CI: 12.5, 28.4) in 2019 (Appendix, Table A1). Overall, MMR in CA countries decreased by 38% from 47.3 (95%CI:29.01, 65.5) per 100,000 live births in 2000 to 29.5 (95%CI: 15.8, 43.1) per 100,000 live births in 2020. In 2020, the MMR in CA countries increased by 44% in comparison to 2019. It is worthy to note that throughout this period, Kyrgyzstan had the highest MMR among the CA countries. Besides, the fluctuations of MMR have taken place in the previous periods of time. There were increases between 2001-2002, 2004-2005, 2008-2009 and 2013-2014 which were comparable in scale to the increase between 2019-2020.
We also added Table A2 to the Appendix section with the absolute values of MM for the selected countries.
- Changes in MMR may not only be due to changes in risk of death but also due to changes in fertility - I would recommend that this this should be expanded upon in the discussion.
Fertility rates in Kazakhstan, Kyrgyzstan, Uzbekistan and Tajikistan in 2020 were higher than the average fertility rate in the world, and higher than in the countries of the European region. Moreover, there was a steady growth of the fertility rate in all three countries (Kazakhstan, Kyrgyzstan, Uzbekistan) over a twenty -year period 2000-2020 (Kazakhstan, Kyrgyzstan), and in 2012-2020 (in Uzbekistan) [https://data.worldbank.org/indicator/SP.DYN.TFRT.IN?locations=KZ] Maternal mortality rate is related to the incidence of pregnancy, however the growth of the fertility rate in CA countries was more gradual than the increase in the maternal mortality observed in Kazakhstan and Kyrgyzstan, therefore one can assume that other factors contributed to that rise.

Reviewer 3 Report (New Reviewer)
Thank you for the opportunity to review this interesting and important paper. I think it contains novel information that will be of interest to readers of this journal.
I have one major concern that I think must be addressed prior to publication, and that is the lack of acknowledgement that there are often fluctuations in the MMR from one year to the next. For example, Figure 1 shows that in Kyrgyzstan there were increases between 2001-2002, 2004-2005, 2008-2009 and 2013-2014 which were comparable in scale to the increase between 2019-2020. The Kazakhstan data provides stronger evidence of a COVID effect, but for the other three countries I would say the evidence is not strong. We will need several years’ more data before we can draw any conclusions about whether or not the recent increases were due to COVID-19. It is absolutely appropriate to draw attention to the recent increases and make hypotheses about the cause(s), but at the same time it is important to acknowledge that it is too early to be confident about those hypotheses. I suggest that the results, discussion and conclusions sections should be re-drafted to address this point.
My other comments are as follows:
Introduction
Line 32: References 1 and 2 are from 2020, when it was too early to judge the impact of COVID-19 on health performance indicators. I suggest to replace one of these references with something more recent, which describes the impact on health indicators, e.g. excess mortality.
Line 32: I would describe the MMR as “a” key indicator, not “the” key indicator.
Line 37: Write out “Central Asia” in full as it is the first time it appears in the main text.
Line 48: The word “thus” isn’t appropriate – do you mean “conversely”? – because the figures from Georgia are different from the global figures
Lines 51-52: The data about older women being at higher risk of MM don’t seem relevant here: this paper is neither from the CA region nor relevant to the research question. Please either explain the relevance of this sentence more clearly or remove it.
Line 56: Please provide citations for the skilled birth attendance figures
Line 57: Data on skilled birth attendance in Tajikistan can be found in the 2017 DHS report (https://www.dhsprogram.com/publications/publication-fr341-dhs-final-reports.cfm )
Lines 65-70: I am surprised that you did not cite Chmielewska et al (2021) here: https://www.thelancet.com/journals/langlo/article/PIIS2214-109X(21)00079-6/fulltext#:~:text=Interpretation,resource%20and%20low%2Dresource%20settings. It seems very relevant to your research question and a high quality study.
Line 85: I don’t think “resulting in” is the right linking phrase here – do you mean “yet in fact it was”?
Materials and methods
Lines 102-119: Please provide citations for the various national data sources. Also, are there any publications which attempt to assess the quality/completeness of the data in these sources? If no, this should be stated. If yes, please cite them and summarise their conclusions.
Lines 120-121: Please provide a citation for the GHE database (one is provided later, but it should be here as well)
Line 121: Why did you decide to use THE as the measure, rather than per capita health expenditure or health expenditure as a % of GDP? These other two measures take into account possible confounders such as population growth and fiscal space, so might show a more accurate picture. The decision to use THE should be defended. If you cannot defend it, I suggest you re-run the correlation analysis with the other suggested measures as a sensitivity analysis.
Line 125-127: I think the reader needs more explanation of the method used to assess the association between MMR and THE. In particular, did you compare the observed MMR figures with THE, or did you use the modelled estimate? If you used the modelled estimate, this decision should be defended.
Results
I think it would be useful context to show the absolute numbers of maternal deaths for each country, as well as the MMR. Small absolute numbers can result in large fluctuations in the MMR and may help to explain the observed variations in the MMR.
Lines 132-134: It would be more accurate to say that Figure 1 shows a general downward trend in all four countries, but fluctuation year-on-year.
Line 140: The trend indicates an average annual decline of 1.31 in the MMR. This is not the same as a decline of 1.31 per year. Same comment applies to lines 153 and 162
Lines 172-179: I am not sure what this analysis adds to the paper. It might be more useful if the THE data covered the same time period as the MMR data, but as the THE data ends in 2019 it doesn’t really tell us much that is relevant to COVID-19.
Table 1: Would the authors consider converting the national currencies to a single currency to aid cross-country comparison?
Line 189-190: Is it possible to unpack the data on indirect causes to see if any individual indirect causes were particularly relevant? The very large number of indirect causes in 2020 suggests that COVID played a major role, but it would be better if the data showed this, rather than leaving it up the reader to assume it.
Figure 3: Can the authors provide any explanation for the relatively large number of ‘others’ in 2020?
Figure 3: When working with such small absolute numbers, it is inappropriate to present %s to 2 decimal places – this implies a level of precision that is neither warranted nor helpful. I suggest just to present the %s as whole numbers.
Figure 3: Were there really zero deaths from indirect causes in 2000, or was it just that they weren’t counted (or counted under ‘other’)?
Lines 196-201: It is not clear why these data are relevant to the research question. The decrease in the number of ob/gyns and maternal beds occurred alongside a trend of decreased MMR, which indicates that these decreases did not have a detrimental effect on maternal mortality. They therefore do not support the point you make in the discussion about under-resourcing being a contributor to MMR
Discussion
Line 211: This statement is not true. The MMR has been on a general downward trend, but 2020 was not the only year in which an increase was recorded.
Line 227-230: I’m not sure how relevant the Amorim reference is to CA, which has low fertility and high maternal health care coverage in comparison to other low- and middle-income countries
Is it possible that the changes over time in causes of maternal deaths in Kazakhstan could be (at least partly) due to changes in the way deaths are recorded and/or classified? e.g. in many countries maternal suicides have only recently been counted as maternal deaths – in fact I wonder if this might explain some of the large overall increase between 2019 and 2020? If so, this should be noted in the conclusions.
Appendix
It would be helpful context if the total fertility rate could be shown as well
It would be more informative if the 5th column showed the % of women accessing prenatal care
Author Response
Response to Reviewers
The authors thank the reviewers for the time and efforts spent to improve the manuscript. All the comments are really appreciated. We agree in general with all points raised by the reviewers and have revised the manuscript considering the suggestions. All of them were very insightful and inspirational.
We hope that the revised manuscript may be contributing to the journal’s interests and values.
The text below follows all the comments from the reviewers. The reviewer´s text is in black and authors´ replies are in red (same as updates and changes in the revised manuscript).
Response to Reviewer 3:
- I have one major concern that I think must be addressed prior to publication, and that is the lack of acknowledgement that there are often fluctuations in the MMR from one year to the next. For example, Figure 1 shows that in Kyrgyzstan there were increases between 2001-2002, 2004-2005, 2008-2009 and 2013-2014 which were comparable in scale to the increase between 2019-2020. The Kazakhstan data provides stronger evidence of a COVID effect, but for the other three countries I would say the evidence is not strong. We will need several years’ more data before we can draw any conclusions about whether or not the recent increases were due to COVID-19. It is absolutely appropriate to draw attention to the recent increases and make hypotheses about the cause(s), but at the same time it is important to acknowledge that it is too early to be confident about those hypotheses. I suggest that the results, discussion and conclusions sections should be re-drafted to address this point.
Thank you for the suggestion.
We addressed this issue in the results and conclusion parts
My other comments are as follows:
Introduction
Line 32: References 1 and 2 are from 2020, when it was too early to judge the impact of COVID-19 on health performance indicators. I suggest to replace one of these references with something more recent, which describes the impact on health indicators, e.g. excess mortality.
We appreciate the suggestion and several references have been changed.
…The COVID-19 pandemic has had a significant impact on maternal mortality in many countries around the world. The pandemic has disrupted essential healthcare services, including maternal and newborn care, leading to an increase in maternal mortality.
- Moynihan R, Sanders S, Michaleff ZA, et alImpact of COVID-19 pandemic on utilisation of healthcare services: a systematic reviewBMJ Open 2021;11:e045343. doi: 10.1136/bmjopen-2020-045343
- Chmielewska B, et al. Effects of the COVID-19 pandemic on maternal and perinatal outcomes: a systematic review and meta-analysis The Lancet Global Health, 2021, Volume 9, Issue 6, e759 - e772. org/10.1016/S2214-109X(21)00079-6
Line 32: I would describe the MMR as “a” key indicator, not “the” key indicator.
We have changed it from “the” key indicator to “a” key indicator.
Line 37: Write out “Central Asia” in full as it is the first time it appears in the main text.
We added “Central Asia” to the sentence.
Line 48: The word “thus” isn’t appropriate – do you mean “conversely”? – because the figures from Georgia are different from the global figures
Thank you for spotting the idea. Yes, indeed, we meant “conversely” (changed it).
Lines 51-52: The data about older women being at higher risk of MM don’t seem relevant here: this paper is neither from the CA region nor relevant to the research question. Please either explain the relevance of this sentence more clearly or remove it.
We agree with you. The sentence is removed.
Line 56: Please provide citations for the skilled birth attendance figures
We fixed the missing citation. This information was found from the SDG Target 3.1 Maternal mortality /www.who.int/data/gho/data/themes/topics/sdg-target-3-1-maternal-mortality.
Line 57: Data on skilled birth attendance in Tajikistan can be found in the 2017 DHS report (https://www.dhsprogram.com/publications/publication-fr341-dhs-final-reports.cfm )
Thank you very much for providing us with a link.
95% of births are delivered by skilled providers in Tajikistan (2017). We added it to the paragraph.
Source: Statistical Agency under the President of the Republic of Tajikistan, Ministry of Health - MOH/Tajikistan, and ICF. 2018. Tajikistan Demographic and Health Survey 2017. Dushanbe, Tajikistan: SA/Tajikistan, MOH/Tajikistan, and ICF. Available at http://dhsprogram.com/pubs/pdf/FR341/FR341.pdf.
Lines 65-70: I am surprised that you did not cite Chmielewska et al (2021) here: https://www.thelancet.com/journals/langlo/article/PIIS2214-109X(21)00079-6/fulltext#:~:text=Interpretation,resource%20and%20low%2Dresource%20settings. It seems very relevant to your research question and a high quality study.
We absolutely agree with you that the Chmielewska article is highly relevant. We fixed our paragraph.
Line 85: I don’t think “resulting in” is the right linking phrase here – do you mean “yet in fact it was”?
We agree with you. The phrase is changed to “yet in fact it was”.
Materials and methods
Lines 102-119: Please provide citations for the various national data sources. Also, are there any publications which attempt to assess the quality/completeness of the data in these sources? If no, this should be stated. If yes, please cite them and summarise their conclusions.
We added the citations for the national data sources (added to the reference list).
For Kazakhstan: Kazakhstan Ministry of Health. Statistical industry data https://www.gov.kz/memleket/entities/dsm/activities/directions?lang=en
For Kyrgyzstan: Center of eHealth of Ministry of Health Republic of Kyrgyzstan http://cez.med.kg
For Uzbekistan: United Nations Population Fund. Uzbekistan https://uzbekistan.unfpa.org/en
For Tajikistan: Agency of Statistics under President of the Republic of Tajikistan https://www.stat.tj/en/news/publications/demographic-yearbook-of-the-republic-of-tajikistan
And added following lines to the “Discussion” section:
“There are few publications which attempt to assess the quality and completeness of the data sources in the CA region. All countries have the eHealth centers that collect and publish annual health statistics information. The legibility of data is considered good in Kazakhstan and Kyrgyzstan. Laatikainen T, Inglin L, Chonmurunov I, Stambekov B, Altymycheva A, Farrington JL. National electronic primary health care database in monitoring performance of primary care in Kyrgyzstan. Prim Health Care Res Dev. 2022 Feb 3;23:e6. doi: 10.1017/S1463423622000019,
Obermann K, , et al. Data for development in health: a case study and monitoring framework from Kazakhstan. BMJ Glob Health. 2016 Apr 18;1(1):e000003. doi: 10.1136/bmjgh-2015-000003
There are quality councils in Kyrgyzstan which have little support in terms of tools, statistical benchmarks or local patient-based data for systematic peer review and improvement. United Nations experts identified the need for capacity development support in implementing measures to manage quality, and to raise awareness of the importance of quality management throughout the national statistical system [World Health Organization. https://www.euro.who.int/__data/assets/pdf_file/0004/383890/kgz-qoc-eng.pdf, https://www.efta.int/sites/default/files/images/publications/GA%20Kyrgyzstan%20Final%20EN.pdf] The efficiency of data harvesting and processing could be improved by stimulating electronic data collection [lobal Assessment of the National Statistical System of Tajikistan https://unece.org/DAM/stats/documents/technical_coop/GA_Tajikistan_ENG.pdf]
Lines 120-121: Please provide a citation for the GHE database (one is provided later, but it should be here as well)
We added it to the line
https://apps.who.int/nha/database
Line 121: Why did you decide to use THE as the measure, rather than per capita health expenditure or health expenditure as a % of GDP? These other two measures take into account possible confounders such as population growth and fiscal space, so might show a more accurate picture. The decision to use THE should be defended. If you cannot defend it, I suggest you re-run the correlation analysis with the other suggested measures as a sensitivity analysis.
It is a great question. We considered that if the “Total health expenditure is the sum of public and private health expenditure”,we believe that it covers the population growth and fiscal space. THE in a single country could be a measure to track if a country has an increase in spending in healthcare even expressed in national currency units (the currency values expressed in USD/Euro would not describe the real picture, since most CA currencies were not reflecting the actual value of the currency for the last decades). We also tried to avoid comparing THE among the countries which would be incorrect to do without converting it to common currency.
Line 125-127: I think the reader needs more explanation of the method used to assess the association between MMR and THE. In particular, did you compare the observed MMR figures with THE, or did you use the modelled estimate? If you used the modelled estimate, this decision should be defended.
We compared the observed MMR figures with THE to measure the strength of the relationship between the two variables. We used the following code:
Correlation <- cor(data$x, data$y, method = 'pearson')
Checking the results: > correlation
Results
I think it would be useful context to show the absolute numbers of maternal deaths for each country, as well as the MMR. Small absolute numbers can result in large fluctuations in the MMR and may help to explain the observed variations in the MMR.
We put the absolute values of MMR in Appendix to Table A2.
Lines 132-134: It would be more accurate to say that Figure 1 shows a general downward trend in all four countries, but fluctuation year-on-year.
We appreciated the suggestion and used it in the line.
Line 140: The trend indicates an average annual decline of 1.31 in the MMR. This is not the same as a decline of 1.31 per year. Same comment applies to lines 153 and 162
Thank you for spotting an important point we missed! We added the updated text to the lines 140, 153 and 162.
Lines 172-179: I am not sure what this analysis adds to the paper. It might be more useful if the THE data covered the same time period as the MMR data, but as the THE data ends in 2019 it doesn’t really tell us much that is relevant to COVID-19.
We found out that there is a strong relationship between the MMR and THE in CA countries and it is in line with what Chen et al identified in their study. https://doi.org/10.1186/s12889-021-11557-3
We took the variables till 2019, because if there were no “corrupted/affected 2020 year”the strong relationship between them would have continued. In addition, by the time we submitted the manuscript we did not have the 2020, 2021 THE values in CA countries.
Table 1: Would the authors consider converting the national currencies to a single currency to aid cross-country comparison?
In order to have a cross-country comparison it is a great suggestion, however the currency values expressed in USD/Euro would not describe the real picture, since most CA currencies were not reflecting the actual value of the currency for the last decades.
We ask you to let us leave the table without changes.
Line 189-190: Is it possible to unpack the data on indirect causes to see if any individual indirect causes were particularly relevant? The very large number of indirect causes in 2020 suggests that COVID played a major role, but it would be better if the data showed this, rather than leaving it up the reader to assume it.
It would be really interesting to unpack the indirect causes, however we did not have access to the detailed data on that. Indeed, as our officials report, most of the 2020 cases were associated with the COVID-19. We hope that in future our next step will be analysis of the COVID-19 associated MM cases in the CA region.
Figure 3: Can the authors provide any explanation for the relatively large number of ‘others’ in 2020?
This data, particularly “other causes”, were derived from the annual statistical packs, and unfortunately, we could not unpack the “other” causes.
Figure 3: When working with such small absolute numbers, it is inappropriate to present %s to 2 decimal places – this implies a level of precision that is neither warranted nor helpful. I suggest just to present the %s as whole numbers.
We followed your suggestion and changed the decimals places.
Figure 3: Were there really zero deaths from indirect causes in 2000, or was it just that they weren’t counted (or counted under ‘other’)?
Indeed, the zero deaths from indirect causes were put/counted under “other causes” in the statistical pack in 2000 in Kazakhstan. If we had a more accurate database from that period we would have presented the more clear information. Unfortunately, we can not. Sorry for that!
Lines 196-201: It is not clear why these data are relevant to the research question. The decrease in the number of ob/gyns and maternal beds occurred alongside a trend of decreased MMR, which indicates that these decreases did not have a detrimental effect on maternal mortality. They therefore do not support the point you make in the discussion about under-resourcing being a contributor to MMR
We thought that information about the decreasing density of gynecologists, maternal beds would be useful for some authors, who need this kind of information, because it is not fully available in English.
We agree that it does not support the point about the under-resourcing, so in the discussion , we make a point that “We believe that under-resourcing in maternal care could have had influence on the increased MMR in some CA countries”. However, we do not associate it with the indicators mentioned above. We believe that maternal care quality is mainly built on investments in medical facilities, workforce capacity, and evidence-based practice implementation.
Discussion
Line 211: This statement is not true. The MMR has been on a general downward trend, but 2020 was not the only year in which an increase was recorded.
Thank you for pointing out our mistakes. We have changed the paragraph as following:
According to the results of our study, the MMR in Kazakhstan and other CA countries has been decreasing annually on average for the last twenty years. In the year of the pandemic there was an increase of MMR in all CA countries, except for Uzbekistan.
Line 227-230: I’m not sure how relevant the Amorim reference is to CA, which has low fertility and high maternal health care coverage in comparison to other low- and middle-income countries
Dear reviewer, here we ask to leave this part in the paragraph. We added that CA countries have:
Fertility rates in Kazakhstan, Kyrgyzstan, Uzbekistan and Tajikistan in 2020 were higher than the average fertility rate in the world, and higher than in the countries of the European region. [https://data.worldbank.org/indicator/SP.DYN.TFRT.IN?locations=KZ]
The THE varies across the CA countries. There are increased investments done in this field but still the level is incomparable to the middle-income countries from other parts of the world. Currently, only Tajikistan has achieved the level recommended by WHO (7% and more of GDP spent on healthcare)
Is it possible that the changes over time in causes of maternal deaths in Kazakhstan could be (at least partly) due to changes in the way deaths are recorded and/or classified? e.g. in many countries maternal suicides have only recently been counted as maternal deaths – in fact I wonder if this might explain some of the large overall increase between 2019 and 2020? If so, this should be noted in the conclusions.
We believe that it is unlikely that the changes over time in causes of maternal deaths can be explained by the changes in the way deaths are recorded and classified, since there were no such modifications within the specified time period.
Appendix
It would be helpful context if the total fertility rate could be shown as well
We added TFR to Table A2 in the Appendix.
It would be more informative if the 5th column showed the % of women accessing prenatal carethe data have limitations and this is not spelled out
We removed the absolute values and put the % of women accessing prenatal care.
We are very grateful for your valuable comments and suggestions.

Round 2
Reviewer 2 Report (New Reviewer)
Many thanks for your thorough response to the earlier review. I congratulate the authors for substantially strengthening this manuscript.
I would just like to flag that part of one of my earlier comments seems to have been missed by mistake, namely: In addition, the specifics of the groups used are currently not clear to me as they are similar to but not the same as the ICD. Within the ICD, ectopic pregnancy (O00) falls within Group 1, Pregnancy with abortive outcome; the various codes for uterine rupture fall within Group 3, Obstetric haemorrhage; but in both these cases these groups also appear separately. It's unclear what 'Other' would include - there's an important difference between cause unknown and another cause such as Complications of anaesthesia. I'd suggest adding a few additional sentences to methods explaining this classification, and including a table as an appendix providing a breakdown of how the ICD codes were grouped for each category.)
Otherweise, congratulations on an great paper.
Author Response
Dear Reviewer,
We are grateful for your recommendations.
Corrections were done in the part “3.2 The comparison of the MM causes before and during the pandemic in Kazakhstan” and stated as follows:
“In the Figure 3, the main causes of maternal mortality are provided. In Kazakhstan, during 2000-2020 years, the main reasons of maternal mortality were broken down to the following categories according to ICD: indirect causes (O10, O24, O98, O99), haemorrhage (O67, O46, O44, O45, O72), abortion (O03-O07), hypertensive disorders (O10-O16), sepsis (O85), uterine rupture (O71.1), ectopic pregnancy (O00), other (O21.1, O22, O71, O73, O87, O88, O90).”
This manuscript is a resubmission of an earlier submission. The following is a list of the peer review reports and author responses from that submission.
Round 1
Reviewer 1 Report
This study examines trends and causes of maternal mortality in Central Asian countries from 2000 to 2020 to examine trends before COVID and effect of COVID. The overall trend is improvement adversely affected by COVID and most prominently for Kazakhstan.
Given the challenges of obtaining accurate and complete maternal mortality data in many countries, the authors might have included discussion of evidence in these countries for complete ascertainment of maternal deaths and how these countries verify cause(s) of deaths.
Reviewer 2 Report
The study was about the trend of maternal mortality rates (MMR) in Central Asia from 2000 to 2020 using a time series and comparison of maternal mortality causes in Kazakhstan before and during the pandemic. A few comments were made to improve the content of this study as below:
1. The associations between the national MMR and CA total health expenditures in NCU were not included in the aims of the study in the Introduction section and were missing in the abstract and conclusion sections.
2. Figure 1 did not show the trend of MMR. Without a graph, it is hard to look for the trend pattern, whether it increased or decreased. I want to recommend the authors have a graph to explain the trend of MMR graphically instead of the current Figure 1, which only illustrates the data for 2020 only.
3. I could not understand why the authors used linear regression in this time series analysis because regression is used for forecasting purposes.
4. Line 136 “Other important indicators have changed over two decades.” What are the other important indicators? Please rephrase.
5. Line 167 to 171. “Linear regression analysis results of 20 years period of the MMR in Kyrgyzstan showed a coefficient of -1.31 (95%CI: -1.91, -0.70, p-value<0.001) and an intercept of 2671.28 (p<0,001). The trend of the yearly MMR in Kyrgyzstan was -1.31, which indicates a yearly decline of 1.31 maternal deaths per 100,000 live births.”.
6. The statements were not parallel with Figure 4 since the regression is used to forecast the future data, not to explain the real data.
7. Line 184-187; 195-198. Linear regression analysis results. Regression is used to forecast future data, not to explain the real data.
8. Line 205 “The absolute number of THE increased in all CA countries from 2000 to 2019”. Please include the values of THE for the years.
Reviewer 3 Report
Thank you providing the possibility to review this paper. It is very timely and important. I like to underscore that I believe that it is important to publish these data. But I like to recommend revisions in the write-up.
The study is of high relevance for the ongoing new MMR estimates of WHO and I recommend you contact CRESSWELL, Jenny at WHO – she would be very happy (cresswellj@who.int).
I recommend to get an English speaking editor from the field who will help to use the appropriate terminology.
Eg the MMR you report is in fact the maternal mortality Ratio and not the rate. The used terminology of the causes of deaths should be inline with WHO terms, eg. hypertensive disorders and indirect causes (not extragenital)
Abstract: Please revise the sentence “the finding proves” as one study can never prove anything
Introduction: I am not sure how useful the reference to the HAQ score is. The score is not commonly internationally used. To characterize the countries, the GDP and the share of funding for health maybe sufficient.
Please check again the review on what is know regarding COVID and pregnancy as this is fast changing, and also changes with the COVID variant…
I recommend against citing the study from Robertson as this study was not based on an empirical study but modelling which had it value at the beginning of the pandemic as eye opener – but not now anymore.
Methods: do you have any information on the quality of the used data sources? Has there been any assessment to check how complete the data are? This is very important to add.
Also describe in detail how the cause of data is assigned and how reliable the information is judged
Results: here you provide some information where no source is mentioned in methods, eg the number of obstetricians
I wonder about the structure of the results section. First data which are available for all countries should be reported and then maybe the causes?
On overall, the results section can be condensed
Discussion
This part could also be condensed and I do not find the part on GDP convincing. Most important, please add a limitation section, the data have limitations and this is not spelled out